# Digital Twins in Pediatric Infectious Diseases: Virtual Models for Personalized Management

**DOI:** 10.3390/jpm15110514

**Published:** 2025-10-30

**Authors:** Susanna Esposito, Beatrice Rita Campana, Hajrie Seferi, Elena Cinti, Alberto Argentiero

**Affiliations:** Pediatric Clinic, Department of Medicine and Surgery, University of Parma, 43126 Parma, Italy; beatricerita.campana@unipr.it (B.R.C.); hajrie.seferi@unipr.it (H.S.); elena.cinti@unipr.it (E.C.);

**Keywords:** digital twin, pediatric infectious diseases, antimicrobial stewardship, precision medicine, diagnostic innovation

## Abstract

Digital twins (DTs), virtual replicas that integrate mechanistic modeling with real-time clinical data, are emerging as powerful tools in healthcare with particular promise in pediatrics, where age-dependent physiology and ethical considerations complicate infectious disease management. This narrative review examines current and potential applications of DTs across antimicrobial stewardship (AMS), diagnostics, vaccine personalization, respiratory support, and system-level preparedness. Evidence indicates that DTs can optimize antimicrobial therapy by simulating pharmacokinetics and pharmacodynamics to support individualized dosing, enable Bayesian therapeutic drug monitoring, and facilitate timely de-escalation. They also help guide intravenous-to-oral switches and treatment durations by integrating host-response markers and microbiological data, reducing unnecessary antibiotic exposure. Diagnostic applications include simulating host–pathogen interactions to improve accuracy, forecasting clinical deterioration to aid in early sepsis recognition, and differentiating between viral and bacterial illness. Immune DTs hold potential for tailoring vaccination schedules and prophylaxis to a child’s unique immune profile, while hospital- and system-level DTs can simulate outbreaks, optimize patient flow, and strengthen surge preparedness. Despite these advances, implementation in routine pediatric care remains limited by challenges such as scarce pediatric datasets, fragmented data infrastructures, complex developmental physiology, ethical concerns, and uncertain regulatory frameworks. Addressing these barriers will require prospective validation, interoperable data systems, and equitable design to ensure fairness and inclusivity. If developed responsibly, DTs could redefine pediatric infectious disease management by shifting practice from reactive and population-based toward proactive, predictive, and personalized care, ultimately improving outcomes while supporting AMS and health system resilience.

## 1. Introduction

Over the past two decades, medicine has progressively shifted from generalized protocols toward more individualized care models. Precision medicine, fueled by genomics, big data analytics, and machine learning, seeks to tailor diagnostics and therapies to the unique biological characteristics of each patient. Within this landscape, the concept of the digital twin (DT)—a dynamic, virtual representation of a physical system that continuously integrates real-world data—has gained traction in healthcare. Initially developed in engineering and manufacturing, DTs now hold transformative potential in biomedical sciences and clinical practice [1].

A digital twin in medicine functions as a computational replica of an individual patient, capable of integrating multimodal data such as genomic profiles, laboratory values, imaging datasets, wearable sensor inputs, and clinical history. This dynamic model evolves alongside the patient, simulating disease progression, predicting treatment responses, and enabling clinicians to virtually test therapeutic strategies before implementation [2]. Such an approach is particularly appealing in pediatrics, where children’s physiology differs markedly from adults and therapeutic decisions must be adapted to ongoing growth, developmental stage, and age-specific immune responses [3].

Infectious diseases remain a leading cause of morbidity and mortality in children worldwide, despite advancements in vaccines, antibiotics, and supportive care. The pediatric population presents unique vulnerabilities: immature immune systems, variable pharmacokinetics and pharmacodynamics, limited clinical trial data, and higher risks of adverse drug reactions [4]. Furthermore, neonates, infants, and adolescents represent distinct biological subgroups, making standardized treatment protocols often insufficient.

Pediatric infectious diseases also pose challenges for public health systems. Seasonal outbreaks of respiratory syncytial virus (RSV), influenza, or emerging pathogens like SARS-CoV-2 underscore the need for accurate predictive tools to guide prevention and optimize hospital resources [5,6]. Traditional models for disease forecasting and therapy selection often lack the granularity to capture the diversity of pediatric patients. Here, DTs could provide individualized insights into host–pathogen interactions, drug efficacy, and long-term outcomes.

The integration of DTs into pediatric infectious disease management offers several promising applications. By virtually modeling immune responses, DTs may help clinicians anticipate vaccine effectiveness in specific age groups or immunologically vulnerable children [7]. Drug dosing regimens can be tailored by simulating pharmacokinetic (PK) profiles that account for weight, organ function, and genetic predispositions [8]. At the health-system level, aggregated DT data could assist in monitoring outbreaks in pediatric wards, enhancing infection control policies, and informing precision public health interventions [9].

Despite these opportunities, DT adoption in pediatrics is still in its infancy. Barriers include limited availability of longitudinal pediatric datasets, technical challenges in creating models that reflect developmental changes, and ethical concerns related to consent and data privacy for minors [9]. Nevertheless, ongoing interdisciplinary research involving clinicians, data scientists, and engineers is laying the foundation for translating DTs from theoretical potential to clinical reality.

This narrative review explores the role of DTs in pediatric infectious disease management, highlighting their potential to personalize therapies, predict outcomes, and support public health strategies. Ultimately, the aim is to provide clinicians, researchers, and policymakers with a comprehensive overview of how DTs could transform the prevention and treatment of infectious diseases in children, paving the way toward truly individualized pediatric care.

## 2. Methods

### 2.1. Literature Search Strategy

A comprehensive literature search was conducted to identify publications on the use of DTs in pediatric infectious disease management. The search covered January 2000 to June 2025 and included PubMed/MEDLINE, Scopus, Web of Science, and Embase. To capture emerging evidence, supplementary sources such as medRxiv, bioRxiv, and the reference lists of relevant reviews were also screened.

Search strategies combined free-text keywords and Medical Subject Headings (MeSH) related to DT, virtual patient, and in silico model with pediatric terms (children, neonates, adolescents) and infectious disease terms (sepsis, antibiotic, antimicrobial resistance, vaccine, immunization). Boolean operators were applied to refine queries. Only English-language publications were considered, and no date restrictions were imposed to ensure inclusion of both seminal and recent studies.

### 2.2. Eligibility Criteria

Studies were eligible if they addressed the concept, development, or application of digital twins, in silico trials, or computational modeling in healthcare with a pediatric focus or a defined pediatric subgroup. Only studies relevant to infectious diseases—including diagnostics, therapeutics, vaccination, or outbreak modeling—were included. Eligible sources comprised original research, systematic or narrative reviews, and policy or position papers.

Exclusion criteria were studies focused exclusively on adults, topics unrelated to infectious diseases, and publications lacking substantive data (e.g., editorials, commentaries, meeting abstracts).

### 2.3. Study Selection

The initial search yielded 2163 articles. After duplicate removal, two reviewers (BRC and EC) independently screened titles and abstracts, leaving 212 articles for full-text evaluation. Following detailed assessment, 30 studies met the inclusion criteria. Disagreements were resolved through discussion and, when needed, adjudication by a third reviewer (HS).

### 2.4. Data Extraction and Synthesis

For each included study, the following data were extracted: authorship, publication year, country, journal, pediatric population (e.g., neonates, infants, children, adolescents), model type (mechanistic, machine learning-based, or hybrid), and domain of application (e.g., drug dosing, vaccine prediction, outbreak forecasting, hospital resource management).

Findings were synthesized thematically, with emphasis on clinical relevance, translational potential, and reported limitations.

### 2.5. Assessment of Methodological Rigor

Because no standardized appraisal tool exists for digital twin research, methodological quality was evaluated using adapted criteria from computational modeling and in silico trial reviews. Assessment domains included transparency of model assumptions and inputs, type of validation (internal, external, or clinical), pediatric-specific adaptations, and utility for clinical or public health decision-making.

Based on these criteria, studies were qualitatively categorized as having high, moderate, or low methodological rigor.

To illustrate the conceptual framework underpinning the DT approach, Figure 1 summarizes the core architectural components and iterative data flow. This structure highlights how multimodal inputs are processed through computational models to generate clinically actionable outputs, which are continuously refined through a dynamic feedback loop.

Multimodal patient and population data—such as electronic health records (EHRs), laboratory and microbiology results, imaging, wearable sensors, and epidemiological inputs—are integrated within mechanistic, machine-learning, or hybrid models to generate predictions and decision support for diagnosis, treatment optimization, and outbreak preparedness. A continuous feedback loop enables real-time model recalibration as new clinical information becomes available, supporting adaptive and personalized management.

## 3. Digital Twins and Antimicrobial Stewardship in Pediatrics

### 3.1. Patient-Level Antimicrobial Optimization

One of the most consistent findings across the literature is that DTs can transform patient-level AMS by tailoring drug exposure to the unique developmental physiology of children. Traditional dosing regimens in pediatrics are often extrapolated from adult data or rely on weight-based formulas that fail to capture interpatient variability in drug clearance, maturation of organ systems, or dynamic pathophysiological changes during infection [4,8]. DTs overcome these limitations by combining mechanistic PK/PD models with data-driven learning modules that are continuously updated with laboratory values, vital signs, and TDM results [2,4]. An example of a DT applied to AMS, integrating PK/PD modeling and Bayesian TDM, is shown in Figure 2.

The model estimates antimicrobial exposure and target attainment, including the fraction of time above the minimum inhibitory concentration (fT > MIC). Acronyms: PK, pharmacokinetics; PD, pharmacodynamics; TDM, therapeutic drug monitoring; MIC, minimum inhibitory concentration; fT > MIC, fraction of time that free drug concentration remains above the MIC.

This capacity for real-time recalibration is particularly critical for time-dependent antibiotics such as β-lactams, where maintaining free drug concentrations above the minimum inhibitory concentration (fT > MIC) is essential, or for concentration-dependent agents like aminoglycosides, where achieving adequate peak concentrations without nephrotoxicity is the primary challenge [8]. By modeling expected trajectories and adjusting dosing dynamically, DTs reduce the time to target attainment and minimize the risks of both under- and overexposure. Studies simulating in silico trials across large pediatric virtual cohorts further demonstrate that DT-guided regimens could significantly reduce empirical overdosing while maintaining microbiological efficacy [4].

### 3.2. De-Escalation, IV-to-Oral Switch, and Duration of Therapy

Beyond initial dosing, DTs show promise in guiding early de-escalation of broad-spectrum antibiotics. By integrating host-response biomarkers (such as fever resolution or C reactive protein [CRP] decline) with emerging microbiology data, DTs can predict with high confidence whether a narrower agent would provide equivalent coverage [2,3]. This is especially important in pediatrics, where ASP must balance safety margins for high-risk populations (e.g., neonates or immunocompromised children) with the imperative to minimize unnecessary broad-spectrum exposure [3].

Similarly, DTs offer a robust framework for determining the optimal timing for intravenous-to-oral (IV-to-PO) switch, a stewardship intervention known to reduce hospital length of stay and line-associated complications. By simulating absorption dynamics, age-dependent bioavailability, and projected tissue penetration of oral formulations, DTs can provide patient-specific projections of therapeutic adequacy [8]. Such simulations enhance clinician confidence in transition decisions, which are often delayed due to uncertainty.

Perhaps most notably, DTs also provide data-driven estimates of relapse risk under different treatment durations. This ability supports the global stewardship push toward shorter, indication-specific therapy courses, reducing cumulative antimicrobial exposure while maintaining cure rates [2,8].

### 3.3. Diagnostic Stewardship and Host–Pathogen Modeling

A critical element of AMS is aligning antimicrobial decisions with accurate diagnostics. Several studies highlight how DTs can act as fusion platforms for host and pathogen data streams. For example, they can integrate rapid molecular diagnostic results with clinical trajectories to refine early treatment choices within the critical first 48 h [2,3]. In viral infections such as influenza or RSV, DTs incorporating immune-response models can differentiate between viral-driven and bacterial-driven illness, thereby avoiding unnecessary antibiotic initiation [5,6].

This function effectively operationalizes diagnostic stewardship, shifting the clinician’s role from interpreting individual test results to evaluating model-based probability distributions that directly inform therapy. By embedding diagnostics into a continuous predictive framework, DTs reduce reliance on empirical therapy and provide a more rational basis for targeted prescribing.

### 3.4. System-Level Stewardship and Policy Testing

At the health-system level, aggregated pediatric DT data enable the construction of virtual cohorts that serve as policy-testing environments. Hospitals can simulate the impact of new empiric sepsis protocols, formulary restrictions, or stewardship interventions across thousands of modeled trajectories before implementing them in clinical practice [2,4]. Outcomes such as antibiotic days of therapy, spectrum scores, or projected resistance pressures can be quantified in advance, enabling stewardship committees to prioritize policies that optimize both clinical outcomes and resistance mitigation.

This is particularly valuable in pediatric wards, where pathogen prevalence and antimicrobial susceptibilities fluctuate with seasonal epidemics such as RSV or influenza [5,6,9]. By tailoring AMS playbooks to the expected seasonal case mix, DTs enhance preparedness while safeguarding vulnerable subgroups. This “virtual policy sandbox” mirrors the methodology of in silico clinical trials and has the potential to accelerate iterative refinement of AMS strategies [4].

### 3.5. Educational Innovations and Virtual Patients

Beyond direct clinical applications, digital strategies for AMS in pediatrics have also advanced significantly through educational simulations, which provide safe, controlled environments for clinical decision-making without patient risk. Virtual patient (VP) modules, for instance, have been piloted in pediatric residency training programs, where learners are presented with complex cases such as severe pneumonia requiring diagnostic interpretation, antimicrobial dosing, de-escalation of broad-spectrum therapy, and decisions regarding treatment duration. These interactive platforms promote active learning by requiring learners to navigate branching scenarios that mimic real-world uncertainty. Importantly, evaluation studies have demonstrated that knowledge scores improved significantly following VP exposure and, crucially, that these gains were sustained at four-month follow-up, confirming the long-term effectiveness of simulation-based education in reinforcing AMS principles and clinical reasoning [10,11,12]. While not DTs in the strict technical sense, VPs exemplify the pedagogical value of virtualization, fostering adaptive thinking and preparing future clinicians to engage with more advanced stewardship technologies.

Complementing these immersive approaches, practical experience with web-based AMS tools highlights the growing role of digital systems in everyday clinical practice. Antibiogram+, for example, is an interactive digital platform that translates microbiological surveillance data into accessible, user-friendly interfaces for clinicians. In pediatric training contexts, its use has been shown to significantly improve prescribing confidence, with 58% of residents reporting that they felt “often” confident in their antimicrobial decisions when using Antibiogram+, compared to only 15% when relying on traditional print-based antibiograms [11]. This transition from static reference materials to dynamic, context-sensitive platforms underscores a broader shift toward digital stewardship ecosystems.

Together, VP modules and web-based tools represent crucial intermediate steps on the path to fully dynamic DT platforms. By familiarizing clinicians with decision-support environments that integrate data and adapt in real time, these innovations lower barriers to adoption and cultivate digital literacy in stewardship. As such, they not only enhance immediate educational outcomes but also lay the groundwork for the successful integration of DTs into routine pediatric infectious disease management [11,12].

### 3.6. Artificial Intelligence in Antimicrobial Stewardship

To avoid conceptual overlap, it is important to clarify the distinction between mechanistic modeling and AI within DT frameworks. Mechanistic models are grounded in established physiological, pharmacological, or immunological principles and use mathematical representations to simulate biological processes and predict system behavior under defined conditions. In contrast, AI and ML models derive patterns directly from data, providing probabilistic predictions without requiring explicit assumptions about underlying mechanisms. In digital twins, these approaches are complementary: mechanistic models offer interpretability and biological plausibility, while AI enhances adaptability and predictive performance by learning from large, heterogeneous datasets. Hybrid models integrating both methods are increasingly used to leverage the strengths of each while mitigating their limitations.

The broader literature on AI in AMS highlights the diverse ways in which ML models can enhance stewardship activities, including inappropriate prescription detection, antimicrobial resistance prediction, and optimization of therapeutic choices [13,14]. These systems leverage large datasets drawn from electronic health records (EHRs), microbiology laboratories, and pharmacy records, applying algorithms to identify prescribing patterns and forecast outcomes that may be difficult for clinicians to recognize unaided. Reported predictive performance across studies is strong, with area under the curve (AUC) values ranging from 0.64 to 0.99, underscoring their reliability in real-world applications [13,14].

ML models have been successfully applied to several specific AMS tasks. For example, natural language processing (NLP) algorithms have been used to extract clinical features from free-text clinical notes, enabling more accurate identification of inappropriate prescriptions. Similarly, predictive models incorporating local resistance data can generate patient-specific antibiograms that recommend the most effective empiric therapy, reducing unnecessary broad-spectrum use. Other approaches focus on early identification of high-risk patients for resistant infections, using demographic, clinical, and laboratory features to guide empiric therapy selection. Collectively, these tools support clinicians by narrowing therapeutic uncertainty and reducing reliance on generalized prescribing practices [13].

Although AI systems are not as comprehensive as DT frameworks, they provide essential methodological building blocks that are directly transferable to DT-based stewardship. These include data harmonization across heterogeneous sources, real-time updating as new clinical and microbiological results become available, and outcome prediction under varying treatment strategies. By embedding these AI-driven capabilities into the continuous, patient-specific simulations of DTs, stewardship interventions could become more dynamic, precise, and adaptive. Thus, AI in AMS not only offers immediate benefits in clinical decision support but also serves as a critical stepping stone toward the realization of fully operational DT platforms in pediatrics [13,14,15,16]. Table 1 summarizes AI and DT application in AMS.

### 3.7. Integration of Antimicrobial Stewardship and Diagnostic Stewardship

A recent review emphasizes the synergistic potential of antimicrobial and diagnostic stewardship programs (ASPs and DSPs) in pediatrics, highlighting the need for multidisciplinary collaboration and the integration of AI-driven decision support [15]. DTs embody this integration by combining therapeutic and diagnostic streams within a single dynamic framework, aligning with the future vision of unified, digitally enabled pediatric stewardship.

Collectively, the literature establishes DTs as a powerful substrate for pediatric AMS. At the bedside, they optimize dosing, enable safer and earlier de-escalation, guide IV-to-PO transitions, and rationalize therapy durations. At the system level, they provide a testbed for stewardship policies and support outbreak preparedness. Educational and digital support platforms, such as virtual patient modules [12] and Antibiogram+ [11], illustrate the preparatory steps toward DT adoption, while AI models [13] demonstrate the predictive backbone that will likely power DT systems. Emerging reviews [16] reinforce the need to integrate AMS with diagnostic stewardship, an objective directly addressed by DT frameworks.

Although challenges remain—including data quality, pediatric-specific model validation, ethical considerations, and equitable access—the convergence of mechanistic modeling, machine learning, and clinical stewardship imperatives strongly positions DTs as a transformative tool to reduce unnecessary antibiotic exposure, mitigate resistance development, and improve outcomes in children [1,2,4,10,11,12,13,14,15,16].

## 4. Early Recognition and Risk Stratification of Deterioration

Early recognition of clinical deterioration is one of the most promising applications of DTs in pediatric infectious disease management. Children often deteriorate rapidly once compensatory mechanisms fail, particularly in sepsis and septic shock, where minutes can make the difference between survival and adverse outcomes [17]. Traditional pediatric early warning scores (PEWS) rely on thresholds for vital signs and are limited in their ability to integrate complex, multimodal data streams [18].

Recent studies demonstrate that machine learning-based systems outperform PEWS by capturing subtle physiologic patterns preceding overt decompensation. Park et al. developed a deep-learning pediatric early warning system using ward vital signs and showed significantly better discrimination for predicting cardiopulmonary arrest compared with traditional scores [19]. Similarly, Scott et al. validated a model using data available within two hours of hospital arrival to predict progression to hypotensive septic shock in children, achieving strong performance metrics and allowing for earlier intervention [20].

Embedding these predictive algorithms within a DT architecture enables continuous recalibration as new data—laboratory values, culture results, and hemodynamic monitoring—arrive. Instead of static alerts, the DT maintains a dynamic risk profile, simulating potential outcomes under different management strategies. Halpern et al. argue that this bi-directional updating, combined with the ability to quantify uncertainty, makes DTs uniquely suited for acute pediatric care, where rapid changes are common and decisions must often be made under uncertainty [18].

By shifting deterioration recognition from reactive thresholds to proactive trajectory modeling, pediatric DTs could help frontline clinicians escalate earlier, allocate resources more efficiently, and potentially reduce morbidity and mortality in infectious disease emergencies.

## 5. Infection Prevention and Control, Cohorting, and Isolation Strategy

DTs are not only patient-centered but also scalable to the hospital system level, where they can improve infection prevention and control (IPC) strategies. Seasonal surges of viral illnesses such as RSV or influenza put immense strain on pediatric wards, where decisions about isolation, cohorting, and room allocation directly affect both patient safety and operational efficiency [5,6].

ML models have already been used to predict RSV severity at admission, allowing early identification of infants likely to require intensive support [21]. When integrated into a DT system, these predictions can feed into ward-level simulations that balance isolation needs with bed occupancy, optimizing placement decisions in real time. By minimizing both unnecessary isolation and missed high-risk cases, DTs strengthen IPC policies without overburdening limited single-room capacity.

At a broader systems level, emergency-care and hospital DTs have been tested to stress workflows, anticipate resource bottlenecks, and optimize cohorting policies during crises [22,23]. For pediatric ID, this could translate into simulating RSV or influenza outbreak scenarios under varying incidence, testing alternative patient-flow strategies, and projecting needs for PPE, ventilators, or high-acuity beds [22]. Importantly, such simulations allow leadership to preview the trade-offs between infection control and operational efficiency before implementing policies that affect thousands of children.

Thus, pediatric DTs serve as dynamic decision-support engines for IPC, ensuring that infection-control measures are evidence-informed, context-aware, and resilient to the unpredictability of seasonal or emerging pathogens.

## 6. Vaccine and Immunoprophylaxis Personalization

Another frontier for pediatric DTs lies in the personalization of vaccines and immunoprophylaxis, an area of immense clinical and public health importance. Children are not a homogeneous population: their immune responses are influenced by developmental stage, genetic background, comorbid conditions, nutritional status, prior antigenic exposures, and environmental factors. Standard immunization schedules, while effective at a population level, are designed for “average” responses and therefore do not account for the wide interindividual variability observed in pediatric populations. This limitation can result in suboptimal protection for certain subgroups, such as preterm infants, immunocompromised children, or those with chronic illnesses.

Laubenbacher et al. have proposed detailed roadmaps for immune digital twins, which leverage mechanistic, multiscale models parameterized with individual-level data—including immune cell counts, cytokine responses, and genetic markers—to forecast vaccine responses [17]. In pediatrics, such immune DTs could help clinicians predict which children are likely to mount durable protective immunity following vaccination, identify those who may require altered booster timing, and determine candidates who might benefit from adjunctive prophylactic measures such as monoclonal antibodies. This capacity for individualized immune profiling could be transformative in managing diseases like respiratory syncytial virus (RSV), where prophylactic monoclonal antibodies are emerging as a complement to traditional vaccines.

Li et al. further emphasize that coupling immune DTs with whole-body and age-specific physiologic models can capture the distinct pharmacokinetics and pharmacodynamics of neonates and infants, enhancing predictions of vaccine safety, dose requirements, and immunogenicity [22]. For example, neonates may exhibit delayed or blunted immune responses compared with older children, necessitating adaptive scheduling or alternative prophylactic strategies. By incorporating these developmental variables, DTs can provide clinicians with a dynamic, individualized roadmap for preventive care.

A unique advantage of immune DTs lies in their ability to quantify uncertainty and simulate “what-if” scenarios. These simulations can explore, for instance, the projected effects of administering a new vaccine in immunosuppressed children or delaying booster doses in the context of vaccine shortages. Such outputs can directly support shared decision-making with families, empowering parents and caregivers with evidence-based projections of risks and benefits when considering novel vaccines or prophylactic agents.

Ultimately, vaccine-personalizing DTs have the potential to move pediatric infectious disease management beyond the “one-size-fits-all” paradigm. By integrating developmental immunology, individualized pharmacology, and predictive modeling, these systems could enable precision immunization pathways that dynamically adapt to each child’s biology and clinical context. In doing so, DTs could not only improve vaccine efficacy and safety at the individual level but also enhance population-level protection by reducing under- or over-vaccination in vulnerable groups [17,22].

## 7. Respiratory Support and Ventilation Optimization

Infectious respiratory illnesses such as bronchiolitis, pneumonia, and post-viral acute respiratory distress syndrome (ARDS) remain leading causes of pediatric morbidity, hospitalization, and intensive care utilization worldwide. Management strategies often rely on supportive care, with non-invasive ventilation (NIV) being a cornerstone intervention. While NIV can prevent the need for invasive mechanical ventilation in many children, its success rates are variable, and predicting which patients will deteriorate remains a persistent clinical challenge. Early recognition of NIV failure is crucial, as delayed intubation is associated with worse outcomes, including higher rates of complications and mortality.

Weaver et al. demonstrated how DT approaches applied to adults with acute hypoxemic respiratory failure could mechanistically explain differential responses to NIV [24]. By incorporating patient-specific data such as lung mechanics, gas exchange dynamics, and clinical parameters into individualized computational models, DTs provide a mechanistic understanding of why certain patients respond favorably to NIV while others progress to respiratory failure. Importantly, these models allow clinicians to “test” alternative ventilation strategies in silico—adjusting pressures, oxygen delivery, and timing—before implementing changes in clinical practice.

In pediatrics, this paradigm carries particular promise. DTs could integrate age-specific respiratory physiology, airway resistance, and lung compliance data to create highly individualized simulations. Such models could help optimize ventilation settings for each child, balancing oxygenation and ventilation targets while minimizing the risk of barotrauma or volutrauma. Moreover, DTs could generate dynamic risk profiles that anticipate the likelihood of NIV failure in real time, offering quantitative escalation thresholds consistent with pediatric critical care guidelines.

The implications extend beyond individual patient management. By reducing unnecessary intubations in children who can safely remain on NIV, while simultaneously preventing harm from delayed escalation in those likely to fail, DT-guided approaches could improve outcomes, reduce ICU length of stay, and preserve scarce critical-care resources. At the health system level, aggregated pediatric DT data could also inform predictive models of ventilator demand during seasonal outbreaks of respiratory syncytial virus (RSV) or influenza, strengthening preparedness for surges.

Ultimately, DT-driven respiratory support strategies represent a shift from reactive escalation to proactive, personalized respiratory care. By bridging mechanistic modeling with real-time clinical monitoring, DTs offer the potential to refine decision-making in one of the most resource-intensive domains of pediatric infectious disease management [24].

## 8. Public Health Linkage and Surge Preparedness

Beyond individual hospitals, DTs can also serve as population-level forecasting tools. By combining school absenteeism data, syndromic surveillance, and local testing trends with hospital census data, DTs can simulate the timing and magnitude of pediatric infectious-disease surges.

Li and Zhang describe how emergency-care DTs can be used to optimize resource allocation and scenario planning in crises [23], while Li et al. highlight the role of global learning health systems built on DTs for preventive medicine [22]. Translating these to pediatrics, DTs could quantify how changes in testing availability, vaccination coverage, or prophylaxis campaigns would influence PICU occupancy and ventilator demand across regional health systems.

Such simulations provide actionable insights for hospital networks and policymakers, enabling proactive coordination across pediatric services and better preparedness for pandemics or seasonal epidemics.

## 9. Data Infrastructure, Validation, and Trial-Readiness

The successful deployment of pediatric DTs depends on robust data interoperability, validation frameworks, and ethical governance. Standardized infrastructures such as FHIR, SMART-on-FHIR, and the OMOP Common Data Model are increasingly recognized as essential for enabling real-time data exchange, model portability, and multicenter validation [ [25], [26],[27],[28]].

Moreover, because DTs function as clinical interventions, their evaluation requires rigorous trial frameworks. The SPIRIT-AI and CONSORT-AI guidelines provide specific recommendations for designing and reporting trials of AI-driven tools, including digital twins [29,30]. These emphasize not just performance metrics but also patient-centered outcomes, bias assessment, and transparent reporting of models-in-operation.

Halpern et al. highlight the importance of uncertainty quantification, governance, and equity safeguards in critical-care DTs, concerns that are magnified in pediatrics where physiology changes rapidly and data are sparser [18]. Ensuring fairness, protecting data privacy, and involving families in decision-making will be central to translating DTs into clinical practice.

Ultimately, building trustworthy pediatric DTs will require not only technical advances but also cultural and ethical frameworks that align innovation with patient safety and equity.

## 10. Future Perspectives

To provide a clearer roadmap for translation, it is useful to distinguish between short-term, feasible applications and long-term, aspirational goals of pediatric DTs. In the near term, the most realistic opportunities include decision-support tools for antimicrobial dosing, integration of DT modules into sepsis early-warning systems, and hospital-level outbreak simulations—areas where data streams and computational models already exist and pilot systems are emerging. By contrast, long-term aspirations such as fully dynamic immune DTs, real-time closed-loop ventilatory control, and global learning-health networks require major advances in data availability, interoperability, regulatory frameworks, and ethical governance. Separating these trajectories highlights what can be implemented within current clinical and technical constraints while maintaining a forward-looking vision for the transformative potential of mature DT ecosystems. Several barriers currently limit the clinical adoption of DTs in pediatric infectious diseases. Table 2 summarizes the key technical, ethical, and regulatory challenges, together with potential solutions to facilitate safe and effective implementation.

The future of DTs in pediatric infectious disease management is compelling, but significant barriers remain before these tools can be fully realized in clinical practice. Table 3 summarizes applications of DTs in pediatric infectious disease management, whereas Table 4 reports the benefits and limitations of DT implementation in pediatrics.

At present, the majority of studies are conceptual frameworks, simulation experiments, or retrospective analyses, with very few prospective trials involving children. This lack of real-world validation raises questions about the reliability, safety, and generalizability of DT-guided decisions across diverse pediatric populations. Data scarcity is a central obstacle: pediatric cohorts are typically smaller, age ranges are heterogeneous, and neonates—who are among the most vulnerable to infectious diseases—are particularly underrepresented. Furthermore, the developmental changes in immune function, organ maturation, and pharmacokinetics make model construction more complex than in adults, and underscore the need for continuous recalibration across the pediatric lifespan. Beyond technical hurdles, practical and ethical challenges must be addressed. Integrating DTs into fragmented health information systems is difficult, especially when real-time interoperability and low-latency data processing are prerequisites. FHIR and OMOP offer promise, but widespread adoption is uneven. Regulatory frameworks for AI-driven clinical tools remain underdeveloped, leaving uncertainty around approval pathways, liability, and long-term governance of in silico decision support systems. Ethical considerations are magnified in pediatrics: questions of parental consent, child assent, data ownership, and privacy must be resolved to ensure trust and fairness. Additionally, biases in training data—particularly the under-representation of children from low-resource settings or those with complex medical conditions—pose risks of inequitable care if not systematically addressed. Despite these limitations, progress is accelerating.

Table 5 summarizes future directions for pediatric DTs in infectious diseases.

Advances in federated learning could enable secure pooling of pediatric data across institutions without breaching privacy, thereby expanding model generalizability. Hybrid approaches that combine mechanistic modeling of physiology with machine learning are emerging as particularly promising, as they harness the interpretability of mechanistic models while benefiting from the adaptive power of AI. Furthermore, DTs are increasingly recognized not only as clinical decision tools but also as testbeds for antimicrobial stewardship policies, outbreak preparedness, and vaccine personalization—expanding their utility from the bedside to the health system level. To translate this potential into practice, future research should prioritize multicenter prospective trials, rigorous adherence to AI trial reporting standards such as SPIRIT-AI and CONSORT-AI, and active engagement of clinicians, ethicists, and families in the design and evaluation process.

If these challenges are met, DTs could drive a paradigm shift in pediatrics—from reactive, population-based strategies to proactive, individualized care that anticipates deterioration, optimizes therapy, and informs public health response. In doing so, DTs would not only improve outcomes for individual children but also strengthen resilience against emerging infectious threats, ensuring that pediatric care remains both precise and equitable in the digital age.

## 11. Conclusions

DTs represent an emerging paradigm with the potential to transform pediatric infectious disease management by enabling continuous, individualized modeling of each child’s physiology, immune responses, and treatment trajectories. To provide an overview of current use cases and their degree of clinical readiness, Table 6 summarizes the main application domains of digital twins in pediatric infectious diseases together with their validation status.

Evidence to date highlights their utility across multiple domains: optimizing antimicrobial stewardship through personalized dosing and de-escalation strategies, enhancing early recognition of sepsis and clinical deterioration, supporting infection control and hospital resource allocation, guiding vaccine and immunoprophylaxis personalization, and providing powerful in silico platforms for policy testing and educational training.

Despite these advances, translation into routine pediatric practice remains limited. Current DT applications are predominantly conceptual, retrospective, or confined to single centers, and the paucity of large, diverse pediatric datasets constrains their generalizability. Integration into fragmented health information infrastructures, ethical and legal challenges surrounding consent and privacy, and the lack of clear regulatory pathways further complicate implementation. Addressing these limitations will require not only technical innovation but also the establishment of rigorous validation frameworks, adherence to reporting guidelines, and multidisciplinary collaboration that actively includes clinicians, ethicists, policymakers, and families.

Looking forward, the convergence of mechanistic modeling, machine learning, and interoperability standards—together with federated data sharing and robust governance—positions DTs as a cornerstone of precision pediatric medicine. If developed responsibly and evaluated rigorously, DTs could shift pediatric infectious disease management from reactive care to proactive, predictive, and personalized interventions, improving outcomes for children while advancing global preparedness against infectious threats.

## Figures and Tables

**Figure 1 jpm-15-00514-f001:**
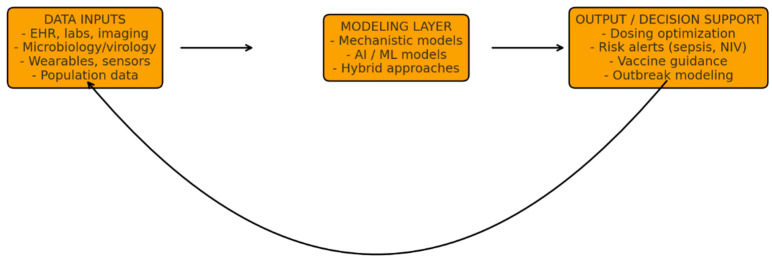
Digital Twin Architecture for Pediatric Infectious Diseases.

**Figure 2 jpm-15-00514-f002:**
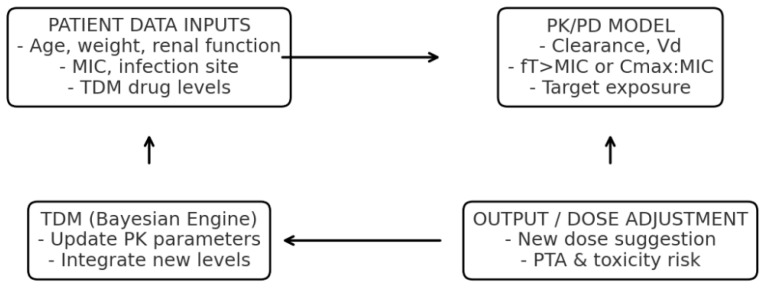
Example of An Antimicrobial Stewardship Digital Twin Integrating Pharmacokinetic/Pharmacodynamic (PK/PD) Modeling with Bayesian Therapeutic Drug Monitoring (TDM).

**Table 1 jpm-15-00514-t001:** Comparison of Artificial Intelligence and Digital Twin Applications in Antimicrobial Stewardship.

Dimension	Artificial Intelligence (AI)/Machine Learning (ML)	Digital Twins (DTs)
Primary Role	Decision support through pattern recognition, prediction, and classification [13,14]	Continuous, individualized simulation of patient physiology, disease progression, and therapy outcomes [2,4,8]
Applications in AMS	• Detection of inappropriate prescriptions • Prediction of antimicrobial resistance • Optimization of empiric therapy choices • Automated extraction of clinical data (e.g., via NLP) [13,14]	• Personalized dosing based on PK/PD models • Bayesian therapeutic drug monitoring • Early de-escalation and IV-to-oral switch guidance • Treatment duration optimization [2,4,8]
Strengths	• Strong predictive performance (AUC 0.64–0.99) • Handles large heterogeneous datasets • Rapid, scalable integration into EHR workflows [13,14]	• Patient-specific, mechanistic modeling • Dynamic recalibration with new clinical/lab data • Simulates outcomes under multiple therapeutic scenarios • Integrates diagnostic and therapeutic stewardship into a unified framework [2,3,8]
Limitations	• Often “black box” models lacking interpretability • Limited to specific tasks (narrow AI) • Requires high-quality, labeled datasets [13,14]	• Sparse pediatric datasets hinder accuracy • Complexity of age-dependent physiology • High computational and data infrastructure requirements • Limited real-world validation [2,9,16]
Complementarity	Provides predictive components (e.g., risk scores, resistance forecasts) that can feed into DTs	Uses AI/ML methods for real-time updating, uncertainty quantification, and improved scalability
Future Potential	Incremental improvements in prescribing safety and efficiency	Paradigm shift toward proactive, predictive, and personalized AMS in pediatrics [2,4,8]

**Table 2 jpm-15-00514-t002:** Barriers and Solutions for Implementing Digital Twins in Pediatric Infectious Diseases.

Domain	Key Barriers	Proposed Solutions
**Technical**	• Sparse pediatric datasets and limited age-specific reference data • Lack of interoperability between hospital systems • Limited real-world validation and model generalizability • High computational demands	• Promote multicenter data sharing and federated learning • Implement interoperability standards (FHIR, OMOP) • Conduct prospective validation and external benchmarking • Use hybrid modeling to reduce data requirements and improve interpretability
**Ethical**	• Consent/assent challenges in minors • Data privacy and security risks • Algorithmic bias and inequity (under-representation of vulnerable groups) • Limited transparency for families and clinicians	• Age-appropriate consent/assent frameworks and parental co-decision • Strong anonymization, encryption, and data-governance policies • Bias monitoring and inclusion of diverse pediatric populations • Promote explainable DT models and shared decision-making tools
**Regulatory**	• Lack of clear approval pathways for DT-based clinical tools • Ambiguity in clinical liability and accountability • Uncertain reimbursement and adoption pathways	• Align DT evaluation with AI-specific trial standards (SPIRIT-AI, CONSORT-AI) • Develop liability-sharing frameworks involving clinicians, hospitals, and vendors • Establish DT reimbursement models within precision-medicine programs

**Table 3 jpm-15-00514-t003:** Applications of Digital Twins in Pediatric Infectious Disease Management.

Domain	Example Applications	Clinical Benefit	References
**Antimicrobial Stewardship**	Personalized dosing, Bayesian TDM, early de-escalation, IV-to-PO switch, duration optimization	Reduced toxicity, faster target attainment, minimized broad-spectrum exposure	[2,4,7,10,11,12,13]
**Diagnostics & Early Recognition**	Sepsis risk prediction, deterioration alerts, pathogen likelihood estimation, host–pathogen modeling	Faster diagnosis, reduced empiricism, earlier escalation	[16,17,18]
**Vaccines & Immunoprophylaxis**	Immune digital twins, booster timing, monoclonal prophylaxis, vaccine efficacy prediction	Precision immunization, reduced over- or under-vaccination	[15,20]
**Respiratory Support**	Simulation of NIV success/failure, ventilation optimization	Timely escalation, reduced intubation delays	[22]
**System-Level Preparedness**	Outbreak modeling, patient flow optimization, cohorting strategies	Improved hospital efficiency, infection control, surge readiness	[19,20,21]

**Table 4 jpm-15-00514-t004:** Benefits and Limitations of Digital Twin Implementation in Pediatrics.

Aspect	Benefits	Limitations/Challenges	References
**Clinical Care**	Individualized therapy, early risk detection, proactive care	Sparse pediatric datasets, high physiologic variability	[2,4,16,18]
**Operational**	Hospital resource allocation, outbreak simulations	Interoperability barriers, real-time data latency	[20,21,24,25,26]
**Education**	Virtual patients enhance AMS training	Limited to pilot studies; uncertain scalability	[11]
**Research**	In silico trials, policy sandboxing	Few multicenter prospective validations	[4,13]
**Ethics & Equity**	Transparency, fairness, child-centered care frameworks	Bias in training data, consent/assent in minors, regulatory uncertainty	[9,16,27,28]

**Table 5 jpm-15-00514-t005:** Future Directions for Pediatric Digital Twins in Infectious Diseases.

Future Area	Key Developments Needed	Expected Impact	References
**Data Integration**	Adoption of FHIR, OMOP, federated learning	Real-time, multicenter interoperability	[24,25,26]
**Validation**	Prospective multicenter clinical trials	Generalizable, trustworthy DT applications	[27,28]
**Hybrid Modeling**	Combination of mechanistic + ML approaches	Improved accuracy and interpretability	[15,16]
**Global Equity**	Inclusion of low-resource pediatric data, fair AI	Reduced bias, broader applicability	[9,20]
**Clinical Translation**	Embedding DTs in AMS, sepsis protocols, vaccination programs	Tangible improvements in outcomes and stewardship	[2,10,11,12,13,18]

**Table 6 jpm-15-00514-t006:** Applications and Validation Status of Digital Twin Approaches in Pediatric Infectious Disease Care.

Domain	Primary Applications	Current Validation Status	Examples/Evidence Type
**Antimicrobial Stewardship (AMS)**	Personalized PK/PD dosing; Bayesian TDM; optimization of treatment duration and IV-to-oral switch	Early-stage/Simulation-level (limited retrospective validation, no prospective pediatric trials)	In silico PK/PD studies; small cohort decision-support pilots
**Diagnostics & Early Recognition**	Sepsis risk prediction; differentiation bacterial vs. viral disease; early deterioration alerts	Moderate validation (retrospective ML validation; early clinical deployment in select centers)	Retrospective EHR-based models; pilot DT frameworks combining host–pathogen data
**Vaccination & Immunoprophylaxis**	Immune DTs for personalized schedules; response prediction; booster timing	Conceptual/Preclinical (mechanistic immune models; no pediatric clinical trials)	Model-based roadmaps and immunological simulations
**ICU/Respiratory Support**	Optimization of NIV settings; prediction of ventilation failure; virtual testing of strategies	Early-stage (validated in adults; extrapolated/simulated in pediatrics)	Mechanistic lung DTs and acute respiratory models (mostly adult data)
**Epidemiology & Preparedness**	Ward-level outbreak modeling; surge prediction; cohorting and resource allocation	Moderate validation (real-world epidemiology models in hospitals; limited pediatric-specific DTs)	Hospital DTs, outbreak simulations, and system-level models

DTs, digital twins; EHR, electronic health record; ML, machine learning; ICU, intensive care unit; NIV, non-invasive ventilation; PK/PD, pharmacokinetic/pharmacodynamic; TDM, therapeutic drug monitoring.

## Data Availability

No new data were created or analyzed in this study.

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
