# Peer review of "Digital Twins in Pediatric Infectious Diseases: Virtual Models for Personalized Management"

_jpm, 2025, doi:10.3390/jpm15110514_

Round 1

Reviewer 1 Report

Comments and Suggestions for Authors

This manuscript provides a narrative review of digital twin applications in paediatric infectious diseases, highlighting their potential in antimicrobial stewardship, diagnostics, vaccination, respiratory support, and system preparedness. The topic is of potential interest to readers, but some aspects would benefit from refinement.

  1. The methods section could be subdivided into clear subsections to improve navigation and readability.
  2. The distinction between mechanistic modelling and artificial intelligence applications should be made clearer to avoid overlap.
  3. The tables should be explicitly referenced in the text at the relevant points to guide the reader.
  4. In the future directions section, the loose single-sentence paragraph should be integrated into the preceding or following paragraph for better flow.
  5. The future perspectives would be stronger if short-term feasible applications were more clearly separated from long-term aspirational ones.
  6. The manuscript would benefit from professional proofreading to correct language inconsistencies and enhance clarity.

Author Response

This manuscript provides a narrative review of digital twin applications in paediatric infectious diseases, highlighting their potential in antimicrobial stewardship, diagnostics, vaccination, respiratory support, and system preparedness. The topic is of potential interest to readers, but some aspects would benefit from refinement.
Re: Thank you for your comments. We revised the manuscript accordingly.

1.    The methods section could be subdivided into clear subsections to improve navigation and readability.
Re: Done as requested (pp. 3-4).

2.    The distinction between mechanistic modelling and artificial intelligence applications should be made clearer to avoid overlap.
Re: Clarified as recommended (p. 7).

3.    The tables should be explicitly referenced in the text at the relevant points to guide the reader.
Re: Revised as requested (pp. 8, 15, and 16).

4.    In the future directions section, the loose single-sentence paragraph should be integrated into the preceding or following paragraph for better flow.
Re: Revised as suggested (p. 16).

5.    The future perspectives would be stronger if short-term feasible applications were more clearly separated from long-term aspirational ones.
Re: A paragraph has been added as suggested (p. 15).

6.    The manuscript would benefit from professional proofreading to correct language inconsistencies and enhance clarity.
Re: The text has been reviewed by an English mother tongue with appropriate knowledge of the manuscript’s content. 

Reviewer 2 Report

Comments and Suggestions for Authors

This is a timely and conceptually strong review addressing the emerging role of digital twin (DT) technologies in pediatric infectious diseases (PID). The authors clearly recognize the critical challenges of individualized care in children—age-dependent pharmacokinetics, developmental immunology, and ethical restrictions—and they propose digital twins as a unifying computational approach to precision pediatrics.

Please consider adding the following figures and tables to your manuscript.

  • Figure 1: Overview of DT architecture for pediatric infectious diseases (Data input → Modeling → Feedback loop).

  • Figure 2: Example of antimicrobial stewardship DT (PK/PD model linked to Bayesian TDM).

  • Table 1: Applications and validation status of DTs in pediatric care (AMS, diagnostics, vaccination, ICU, epidemiology).

  • Table 2: Barriers and solutions (technical, ethical, regulatory).

Author Response

This is a timely and conceptually strong review addressing the emerging role of digital twin (DT) technologies in pediatric infectious diseases (PID). The authors clearly recognize the critical challenges of individualized care in children—age-dependent pharmacokinetics, developmental immunology, and ethical restrictions—and they propose digital twins as a unifying computational approach to precision pediatrics.
Re: Thank you for your comments. We revised our manuscript accordingly..

Please consider adding the following figures and tables to your manuscript.
•    Figure 1: Overview of DT architecture for pediatric infectious diseases (Data input → Modeling → Feedback loop).
Re: Added as requested (pp. 4-5).
•    Figure 2: Example of antimicrobial stewardship DT (PK/PD model linked to Bayesian TDM).
Re: Added as requested (pp. 5-6).
•    Table 1: Applications and validation status of DTs in pediatric care (AMS, diagnostics, vaccination, ICU, epidemiology).
Re: Added as suggested (pp. 19-20).
•    Table 2: Barriers and solutions (technical, ethical, regulatory).
Re: Added as requested (pp. 16-17).

Round 2

Reviewer 1 Report

Comments and Suggestions for Authors

The authors have successfully addressed my main concerns.

Reviewer 2 Report

Comments and Suggestions for Authors

The authors have successfully covered the reviewer's suggestions.